# Microstructure and Mechanical Properties of Galvanized-45 Steel/AZ91D Bimetallic Material by Liquid-Solid Compound Casting

**DOI:** 10.3390/ma12101651

**Published:** 2019-05-21

**Authors:** Jun Cheng, Jian-hua Zhao, Jin-yong Zhang, Yu Guo, Ke He, Jing-Jing Shang-guan, Fu-lin Wen

**Affiliations:** 1College of Materials Science and Engineering, Chongqing University, Chongqing 400045, China; cj950609@163.com (J.C.); zhangjinyongcqu@163.com (J.-y.Z.); gybee183@163.com (Y.G.); hk7313@hotmail.com (K.H.); happysgjzll@163.com (J.-J.S.-g.); wfl_cqu@163.com (F.-l.W.); 2National Engineering Research Center for Magnesium Alloys, Chongqing University, Chongqing 400044, China

**Keywords:** hot-dip galvanizing, solid-liquid compound casting, intermediate compound, interface microstructure

## Abstract

A connection between hot-dip galvanized 45 steel and AZ91D was achieved by liquid-solid compound casting to achieve one material with a better mechanical performance and a light weight. The microstructure and properties of galvanized-steel/AZ91D bimetallic materials were investigated in this study. A scanning electron microscopy (SEM), an energy dispersive spectroscopy (EDS), and an X-ray diffraction (XRD) were applied to analyze the microstructure evolution and formation mechanism of the galvanized 45 steel/AZ91D interface zone which could be divided into three layers. Among three different layers, the layer close to AZ91D was composed of α-Mg and an eutectic structure (α-Mg + MgZn). The intermediate layer was comprised of an eutectic structure (α-Mg + MgZn), and the layer adjacent to 45 steel consisted of α-Mg and FeAl_3_. Furthermore, galvanized-45 steel/AZ91D bimetallic material had better shear strength than the bare-45 steel/AZ91D metallic material which can indicate that owing to the formation of metallurgical bonding, the adhesive strength of galvanized-steel and AZ91D was improved to 11.81 MPa. In addition, the fact that corrosion potential increased from −1.493 V to −1.143 V and corrosion current density changed from 3.015 × 10^−5^ A/cm^2^ to 1.34 × 10^−7^ A/cm^2^ implied that the corrosion resistance of galvanized-steel/AZ91D was much better than AZ91D.

## 1. Introduction

Magnesium alloys are more and more widely employed in the automobile field because of its lighter weight, better specific strength and higher specific stiffness than most common structure metals [1,2,3]. However, its poor room temperature plasticity and corrosion resistance limit wider applications in modern industry [4,5]. Compared with magnesium alloys, steel has better room temperature plasticity and corrosion resistance. The reliable combination between steel and magnesium alloy was an effective method to achieve steel/Mg bimetallic material with a light weight and better operational performance which would replace the most common structure material in the automobile field. However, the huge differences between Mg alloy and steel lead to the following challenges in joining of steel and magnesium alloy: Firstly, according to the Fe-Mg equilibrium phase diagram [6], the solubility of Fe in Mg is 0.00043 at.%, and the solid solubility of Mg in Fe is close to zero. What is more, there was no intermetallic compound between Mg element and Fe element in the joining of Mg alloy and steel. Secondly, there was also a huge difference in the melting point (922 K (Mg) and 1811 K (Fe)). Thus, it was a challenge to realize the reliable combination between the Mg alloy and steel. Even if so, there are still some interconnection techniques for steel/Mg bimetallic material such as ERW (electric resistance welding) [7,8], FSW (fricition stir welding) [9,10,11], (LW) laser welding [12], diffusion welding [13] and ultrasonic spot welding [14,15]. Among these technologies, there was a common point where an interlayer such as Zn [16,17,18], Ni [19], Al [20], Cu [21] became beneficial to realize steel/Mg metallurgical bonding, and Zn interlayer was one of the best common interlayers in these technologies. For example, Cao et al. [17] used cold metal transfer welding-brazing to produce a galvanized 45 steel/AZ31 bimetallic material. Two interlayers ((α-Mg + MgZn) + Fe-Al IMC) were generated at the Mg/galvanized steel interface and an Fe-Al intermediate compound was formed at the Mg/bare steel interface. And Liu et al. [7] used resistance spot welding to produce AZ31B /hot-dip galvanized HSLA steel dissimilar metals. In addition, ultrasonic spot welding (USW) was applied by Patel et al. [14] to achieve an Mg/galvanized steel bimetallic material and the shear load of the Mg alloy/steel bimetallic material reached 47 MPa. Although these technologies could realize the joining of magnesium alloy and steel, they were still several problems. Firstly, these technologies could not provide a combination of large-scale or complex parts. Secondly, these technologies could not reduce or even eliminate the corrosion of magnesium alloy. Thirdly, these welding procedures lacked joining efficiency and flexibility.

Solid-liquid compound casting is a technology using pouring liquid mental on solid metal to get the connected bimetallic material, which has been applied to fabricate different bimetallic materials such as Fe/Cu [22], Al/Mg [23], steel/Al [24] and Al/Cu [25]. Although the mechanical bond between bare steel and Mg alloy could be achieved by this technology, it was difficult to realize metallurgical bonding and get good mechanical properties. 

In this paper, a novel technology which combined hot-dip galvanizing and solid-liquid compound casting was applied to achieve Fe/Mg metallurgical bonding. Hot-dip galvanizing is one of the most common methods to a realize clad layer which can be worked as an interlayer to facilitate a metallurgical reaction between magnesium alloy and steel and protect the Fe/Mg bimetallic material from corrosion. In the experience, a layer of zinc pseudo-alloy coating was deposited on the 45 steel by hot-dip galvanizing and then the molten liquid Mg alloy was cast into the mold, which was inlayed with the hot-dip galvanized 45 steel to achieve an AZ91D/45 steel bimetallic material. Compared with the welding technologies, this technology could not only realize a combination between Mg alloy and steel, but also played a significant role in improving the corrosion resistance of magnesium alloy. Moreover, the microstructure characterization of hot-dip galvanizing coating and galvanized- steel/AZ91D bimetallic material interface was investigated systematically, and the mechanical and corrosion test was also applied to detect the properties of galvanized-steel/AZ91D bimetallic material.

## 2. Experimental

### 2.1. Materials

The materials employed in this study were AZ91D, 45 steel and pure commercial zinc. For hot-dip galvanizing, the matrix material was 45 steel, the dimension of which was 100 mm × 100 mm × 3 mm, and for liquid-solid compound casting, the matrix material was AZ91D. In addition, RJ-2 flux was chosen to prevent the oxidation of the liquid AZ91D. The chemical composition of the materials was shown in Table 1.

### 2.2. Preparation of AZ91D/45 Steel

In the process of hot-dip galvanizing, the 45 steel was dipped into the pure molten zinc at 500 °C for 120 s. Before hot-dip galvanizing, 45 steel was immersed into the mixed solution of 20% ammonium chloride solution and 20% zinc chloride solution at 80 °C for 10 min, then dried instantly at 120 °C. After that, the galvanized 45 steel, bare 45 steel and the mold were preheated at 200 °C for one hour. At the same time, AZ91D was placed into the graphite crucible in an electrical resistance furnace at the elevated temperature to the 720 °C which was kept for 5 min under the protection of RJ2 covering flux. Finally, the melted AZ91D was casted into the mold inserted the bare or the galvanized steel, respectively. Figure 1 showed schematic diagram of outer cladding by solid-liquid compound casting. What is more, in order to improve the shear strength of AZ91D and 45 steel, the zinc coating was sandblasted by the sandblast mechanical. The grit applied in this experience is the alumina grit, the dimensions of which are 0.7–0.85 mm, while the working pressure was 0.7 MPa.

### 2.3. Characterization

The specimens of 10 mm × 10 mm × 10 mm were cut from the cross-section of the interface by a computer numerical control line cutting machine and were grounded by silicon carbide papers up to 2000 grit. And then the samples were etched by 3 vol.% HNO_3_ in alcohol solution. The microstructure characterizations of samples were observed by TESCANVEGAIII scanning electron microscope (SEM, TESCAN, BrNo, South Moravia, Czech Republic) and the distribution of chemical compositions and the composition of the phrases were analyzed by an energy dispersive X-ray spectrometer (EDS, Oxford instruments, Oxford, UK) detector. In addition, the detailed phase composition analysis of the interface was examined by a D/max 2500PC X-ray diffraction (Rigaku, Tokyo, Japan).

In order to evaluate the corrosion performance of galvanized-steel/AZ91D bimetallic material and AZ91D, corrosion test of the samples were experienced in 3.5% NaCl solution at 25 °C. Actually, the potential in the range of −3.0 V to +0.5 V was applied to the three-electrode system cell (Shanghai chenhua instrument, Shanghai, China) composed of the working electrode, a platinum chip as the counter electrode, and a saturated calomel electrode (SCE) as the reference electrode, and the data was recorded at a scan rate of 5 mV/s.

The microhardness distributions from galvanized 45 steel to AZ91D along the vertical direction of the interface were measured by an Everone MH-3L microhardness tester (Everone, Singapore). Moreover, in order to accurately measure the microhardness, suitable testing load and holding time were used. A testing load of 50 N for a holding time (15 s) were applied in AZ91D substrate and 45 steel.

The shear strength of the interfaces in AZ91D/bare 45 steel and AZ91D/galvanized steel were tested to estimate the bond strength between AZ91D and 45 steel. And in order to achieve a more correct strength, three specimens were used for testing. The shear test equipment P4000A (Island ferry company, Kyoto, Japan) is shown in Figure 2, and the shear strength of the interface could be calculated based on the equation as follows:(1)$$\delta_{\tau }=\frac{F_{max}}{2hk}$$
where $F_{max}$ is the maximum load, *h* is the thickness of specimen, and *k* is the length of specimen.

## 3. Results and Discussion

### 3.1. Analysis of Hot-Dipping Galvanized Coating

The Scanning Electron Micrograph (SEM) image of the cross-section of hot-dip galvanized steel was shown in Figure 3a and the its corresponding EDS line scan was displayed in Figure 3b. It can be seen that the cross-section morphology of zinc coating can be divided into four different layers: the outermost layer A, the layer B, the layer C, the layer D at the 45 steel side as marked in Figure 3a. The average thickness of zinc coating was 28 um, which was measured by EDS scan. According to Figure 3b, from the 45 steel to layer A, the content of Fe element gradually decreased, while the content of C and Zn elements had little change. In order to further illustrate the distribution of Fe, C and Zn elements, the EDS map scan morphology was shown in Figure 4. The result showed that a small number of zinc elements diffused into the 45 steel and a small quantity of the Fe element diffused into the zinc coating to form a (Fe, Zn) binary phase. Compared with the Fe element, the C element diffused adequately and was evenly distributed in all positions.

In order to identify the composition of different layers, the EDS point scan was conducted on a different layer. The results of Fe and Zn elements by point scan were displayed in Table 2.

According to Fe-Zn binary equilibrium phase diagram [26] and the results of Fe and Zn elements by point scan, at the layer D, Τ(Fe_3_Zn_10_) was formed due to $\alpha -\mathrm{Fe}+L\to {\mathrm{Fe}}_{3}{\mathrm{Zn}}_{10}$ peritectic reaction, and at the layer C, δ(FeZn_7_) was generated on the basis of peritectic reaction ($Τ\left({\mathrm{Fe}}_{3}{\mathrm{Zn}}_{10}\right)+L\to \delta ({\mathrm{FeZn}}_{7})$). What was more, because of the decrease of Fe content, ζ(FeZn_13_) was formed at layer B due to peritectic reaction ($\delta \left({\mathrm{FeZn}}_{7}\right)+L\to \zeta ({\mathrm{FeZn}}_{13})$). Finally, the melted Zn (η phase) was solidified on the surface of the ζ phase.

### 3.2. Analysis of Galvanized-Steel/AZ91D

Figure 5 showed SEM micrograph of interface in galvanized-steel/AZ91D and the related EDS line scan morphology. It could be seen that the interface between AZ91D and 45 steel can be divided into three different layers: a layer adjacent to the AZ91D substrate (I), a layer close to the 45 steel (III) and an intermediate layer (II) as marked in Figure 5a. The layer I was mainly composed of the black phase and white phase, the layer II consisted of a white phase and the layer III was comprised of a small block black and white phase.

As can be seen from Figure 5b, from AZ91D to 45 steel, the content of Mg, Al, Zn and Fe elements changed slowly as a whole. But in some positions, the content of Mg, Al and Fe elements changed a great deal. The content of Mg element in the layer II was even, and the content of Zn element in the layer II was uniform and more abundant than the other position. In addition, the peak of content of Al and Fe elements presented to layer III, while the content of Al and Fe elements in other layers is close to zero.

Corresponding EDS maps of the SEM cross-sectioned micrographs of galvanized-steel/AZ91D was showed in Figure 6. The content of Al, Fe and O elements in layer III was more abundant than other layers. In addition, the content of Mg element was rich in black phase. It could be worth noting that the content of Zn element in the white phase was much higher than in other positions.

To better investigate the microstructure of three different layers, the SEM morphology at high magnification was detected and the corresponding result was displayed in Figure 7. As can be seen clearly from Figure 7, the constituents of the three layers differed from each other. Based on the formation of different phases between AZ91D and 45 steel, it could be inferred that the formation mechanism was different and was related to the different temperature. In order to identify these phases, an energy dispersive spectrometer (EDS) point scan was applied, and the result was shown in Table 3.

In order to accurately identify the phases, XRD detection was applied in the center of interface zone and the corresponding result was shown in Figure 8. It could prove the existence of α-Mg, MgZn, Al_12_Mg_17_, Fe and FeAl_3_ in the test area. According to the bind energy of and element content, there were not Al_12_Mg_17_ and Fe in the interface zone. So, it could be deduced that there are α-Mg, MgZn and FeAl_3_ in the interface zone.

Based on the Mg-Zn binary phase diagram [26], EDS point scan and the result of XRD, in the layer close to AZ91D, the white uniform lamellae phase was a (α-Mg + MgZn) eutectic structure which can be generated due to the eutectic reaction, and the black block phase was α-Mg solid solution which contains small quantities of Al and Zn. In the intermediate layer, the white lamellae structure was an (α-Mg + MgZn) eutectic structure which can be generated because of the eutectic reaction. Formation of MgZn even far from the steel surface confirmed that the diffusion of Zn reached the AZ91D substrate. Actually, on the one hand, the formation of MgZn advantages the metallurgical bonding. On the other hand, the existence of a brittle and fracture MgZn phase disadvantaged the shear strength.

According to the energy dispersive spectrometer (EDS) line scan in Figure 6b, the Al element was enriched in the layer III while the content of Zn element in this layer is smaller than other layers. Furthermore, in the layer III, Al element prefers to react with the Fe element instead of the Mg element to form an FeAl_3_ intermediate compound which contained a few Zn atoms, and the block black phase was α-Mg, which contained a few Zn and Al atoms. In order to verify this statement, an EDS map scan was applied in Figure 9. The black blocky phase area contained an Mg element and a small amount of Zn and Al elements, while the small white block phase contained iron, Zn and Al elements without Mg element, which could further prove the result of EDS point scan. The O element was abundant in the layer close to 45 steel due to the oxidation of zinc and magnesium alloy, but the content of the O element was negligibly low.

In order to explain the role of Zn coating, the cross-section image of AZ91D/bare-45 steel and the correspond EDS line scan morphology was displayed in Figure 10. It could be concluded that there is no reaction layer between AZ91D and bare 45 steel. Although there is no reaction layer, mechanical bonding still exists between AZ91D and 45 steel, and it was worth noting that the existence of Al element in the 45 steel can implies that under the influence of a liquid Mg alloy, Al element was diffused into 45 steel. However, the content of the Al element was negligible. In a conclusion, the Zn coating is beneficial to the formation of reaction layers and metallurgical bonding. In other words, Zn coating improves the surface wettability of the steel.

### 3.3. Bonding Mechanism of Galvanized-Steel/AZ91D

At the beginning, wettability should be discussed, which is related to the formation of metallurgical bonding between the solid and liquid metals. According to the Wenzel model [24], increasing the roughness of the Zn surface is beneficial to improving the bond strength between AZ91D and 45 steel. Sandblasting is an effective method to improve the surface roughness. In addition, under the impact of the grit, the oxide layer on the Zn coating was broken and the new surface contacted with the liquid Mg.

Based on the above analysis, the schematic of microstructure evolution in the galvanized-steel/AZ91D interface was showed in Figure 11 below and can be divided into four stages: (1) filling process of liquid metals; (2) diffusion process and chemical reactions; (3) metallurgical reaction among the Al, Fe, Mg, Zn elements; (4) solidification as shown in Figure 11. Firstly, AZ91D liquid metals filled the cavity instantly and when the liquid Mg alloy contacted with the Zn coating, the liquid Mg solidified instantly as shown in Figure 11a. At the second stage was diffusion process and chemical reactions as shown in Figure 11b. Under the heat of liquid Mg alloy, Zn coating reacted with solidified Mg alloy, forming different phases in different layers. During this process, the diffusion process and chemical reactions took place in the interfacial zone. According to the diffusion theory, a lot of Mg and Al diffused into Zn coating and a few Zn diffused into AZ91D substrate. What was more, on account of the thin Zn coating, the Mg and Al diffused adequately under the temperature gradient and concentration gradient. In addition, the content of Mg and Al elements in the layer I was close to the layer II. Because the diffusibility of Zn into Mg was smaller than Mg into Zn, a small quantity of Zn diffused into AZ91D. According to the fact that binding energy of Mg and Zn was smaller than Zn and Fe [16,17,18], when the concentration of the Mg atom exceeded the solid solution limit of Zn instantaneously, the eutectic structure and α-Mg were formed. With the increase of the diffusion distance, the diffusibility of Zn element gradually reduces and the content of Zn element gradually increases. So, when the concentration of the Mg element exceeded the solid solution limit of Zn element, the eutectic structure was generated. The third stage was full of metallurgical reactions among the Al, Fe, Mg, Zn elements (Figure 11c). The Al element reached layer III and reacted with Fe to form a FeAl_3_ phase, and a few Zn dissolved into FeAl_3_. With the formation of FeAl_3_, the concentration gradient of Al element increased and more Al diffused into layer III. At the same time, α-Mg was formed because of solution limits. Finally, the metallurgical interface which consisted of an (MgZn + α-Mg) eutectic structure, α-Mg and FeAl_3_ was generated between 45 steel and AZ91D after solidification (Figure 11d).

To sum up, the formation of three different layers is mainly influenced by several factors: the capacity of Mg into the Zn coating, the capacity of Zn into the AZ91D, binding energy between different atoms, and the reaction among Al, Fe, Zn, Mg elements.

## 4. Corrosion and Mechanical Properties

### 4.1. Corrosion Test

In order to evaluate the corrosion performance of galvanized-steel/AZ91D bimetallic material and AZ91D, the corrosion tests of these specimens were carried out in 3.5 wt.% NaCl solution at 25 °C. Polarization curves of the galvanized-steel/AZ91D bimetallic materials and AZ91D are shown in Figure 12, the coordinate x-axis represents the corrosion potential relative to the saturated calomel electrode (SCE) and the y-axis represents the logarithm of corrosion current density. Compared with AZ91D, galvanized-steel /AZ91D bimetallic material had better corrosion resistance. The corrosion potential of AZ91D was −1.493V, while the corrosion potential of galvanized steel/AZ91D bimetallic material was −1.143V. In addition, the corrosion current density of AZ91D and galvanized-steel /AZ91D bimetallic material were 3.015 × 10^−5^ A/cm^2^ and 1.34 × 10^−7^ A/cm^2^, respectively. These phenomenon indicated that the zinc coating on the surface could effectively improve the corrosion resistance of the galvanized-steel/AZ91D bimetallic material. Actually, hot-dip galvanizing coating on the surface could work as the protective layer due to electric potential difference between 45 steel and Zn. The battery was composed of two kinds of coating layers (hot-dip galvanizing coating and 45 steel substrate). Because hot-dip galvanizing layer has a lower electric potential than 45 steel substrate, hot-dip galvanized coating is oxidized as the anode, while 45 steel substrate is protected as the cathode. Because of the pyknotic ZnO film, the rate of corrosion was slow.

### 4.2. Microhardness Distribution at the Galvanized-Steel/AZ91D Compound Interface

The hardness test is an effective method to represent the resistance of the material to local plastic deformation and destruction. Figure 13 showed the microhardness distributions from galvanized 45 steel to AZ91D along vertical direction of the interface. It can be clearly seen that the average microhardness values of the interface was higher than AZ91D because of the existence of intermediate compound, but was also lower than 45 steel. In addition, average microhardness values of the interface was uneven due to the different phase. Among three layers, the layer close to 45 steel (layer III) reached 325.4HV which was highest than other layers due to the existence of FeAl_3_. In addition, from the layer close to AZ91D (layer I) to the intermediate layer (layer II), the average microhardness values increased from 104.8 HV to 139.3 HV because of the increase of content of MgZn and α-Mg.

### 4.3. Shear Test of 45 Steel/AZ91D

Shear test is one of the most important methods to estimate the bond strength between 45 steel and AZ91D. Table 4 displayed the shear strength of galvanized-steel/AZ91D bimetallic material. The shear strength of bare 45 steel/AZ91D bimetallic material was so bad, it was split during the preparation of shear test samples by computer numerical control line cutting machine. The average shear strength values of galvanized-steel/AZ91D reached 11.81 MPa the variation of which was due to the change of quantity of intermediate compound. The more content of hard and brittle phase MgZn, the lower shear strength of galvanized-steel/AZ91D. Compared with bare 45 steel/AZ91D bimetallic material, the galvanized-steel/AZ91D bimetallic material had better shear strength which implied that Zn coating is beneficial to increase shear strength of steel/AZ91D bimetallic material, because metallurgical bonding replaced mechanical bonding.

## 5. Conclusions

(1)In galvanized-45 steel/AZ91D bimetallic material, based on the existence of the interface zone, the metallurgical bonding between galvanized-45 steel and AZ91D was achieved via pouring the molten magnesium alloy into the mold inserted into galvanized 45 steel. However, there was only mechanical bonding between bare steel and AZ91D via solid-liquid compound casting.(2)The interface zone between galvanized 45 steel and AZ91D could be divided into three different layers. The layer adjacent to the AZ91D (layer I) was mainly composed of a (α-Mg + MgZn) eutectic structure and a black block phase (α-Mg). The layer close to the 45 steel (layer III) was mainly comprised of small white block FeAl_3_ and black block α-Mg, and the intermediate layer (layer II) consisted of a white uniform lamellae phase (α-Mg + MgZn) eutectic structure.(3)The galvanized-45 steel on the surface of galvanized-steel/AZ91D bimetallic material could effectively improve the corrosion resistance of AZ91D, which could be proved by the fact that corrosion potential increased from −1.493 V to −1.143 V and corrosion current density changed from 3.015 × 10^−5^ A/cm^2^ to 1.34 × 10^−7^ A/cm^2^.(4)With the change of the composition in different layers, the microhardness of galvanized-steel/AZ91D bimetallic material varied from location to location. From the layer I to the layer II, the microhardness increased gradually from 104.8 HV to 139.3 HV due to the increasement of MgZn phase contents. But from layer II to layer III, the microhardness changed rapidly from 139.3 HV to 325.4 HV, because the microhardness of the FeAl_3_ was much larger than the MgZn phases.(5)The shear strength of galvanized-steel/AZ91D bimetallic material was much better than bare 45 steel/AZ91D bimetallic material, because of the metallurgical bond replacing the mechanical bond.

## Figures and Tables

**Figure 1 materials-12-01651-f001:**
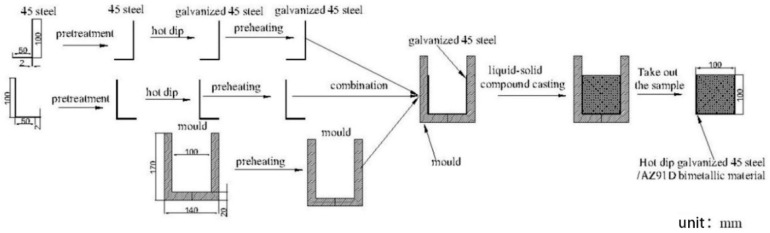
Schematic diagram of outer cladding by solid–liquid compound casting.

**Figure 2 materials-12-01651-f002:**
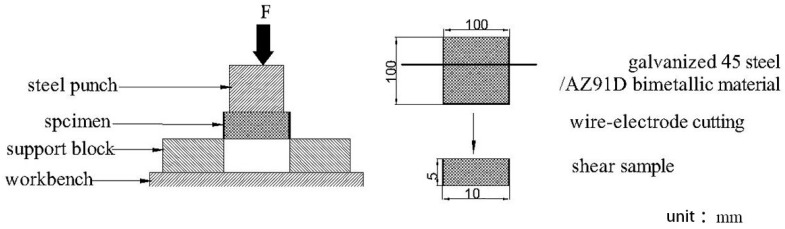
Schematic illustrations of the shear test.

**Figure 3 materials-12-01651-f003:**
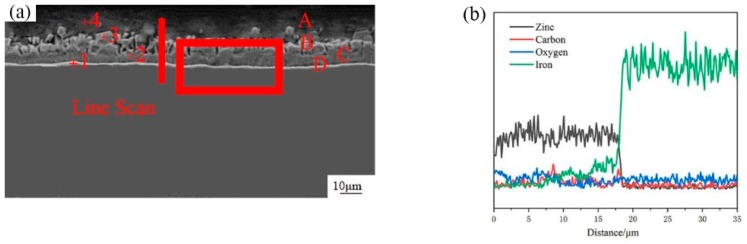
(**a**) SEM micrograph of interfacial microstructure of cross-section image of hot-dip galvanized 45 steel; (**b**) EDS line scan of cross-section of hot-dip galvanized 45 steel marked in (**a**).

**Figure 4 materials-12-01651-f004:**
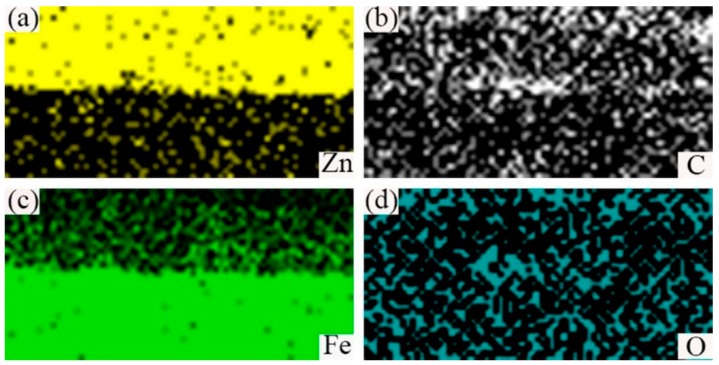
EDS maps scan of the cross-section image of hot-dip galvanized 45 steel marked in Figure 3a. (**a**)EDS map of Zn element; (**b**) EDS map of C element; (**c**) EDS map of Fe element; and (**d**) EDS map of O element.

**Figure 5 materials-12-01651-f005:**
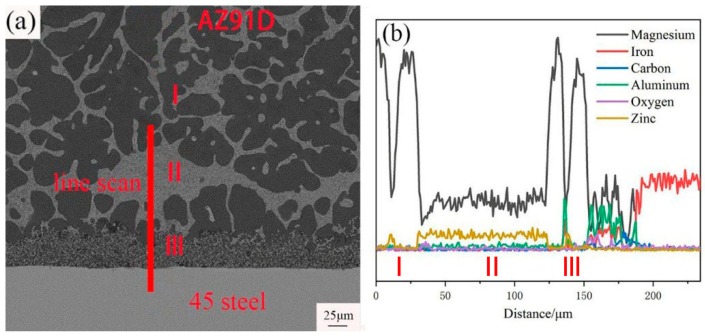
(**a**) SEM micrograph of interface in galvanized-steel/AZ91D; (**b**) EDS line scan of galvanized- steel/AZ91D interface marked in (**a**).

**Figure 6 materials-12-01651-f006:**
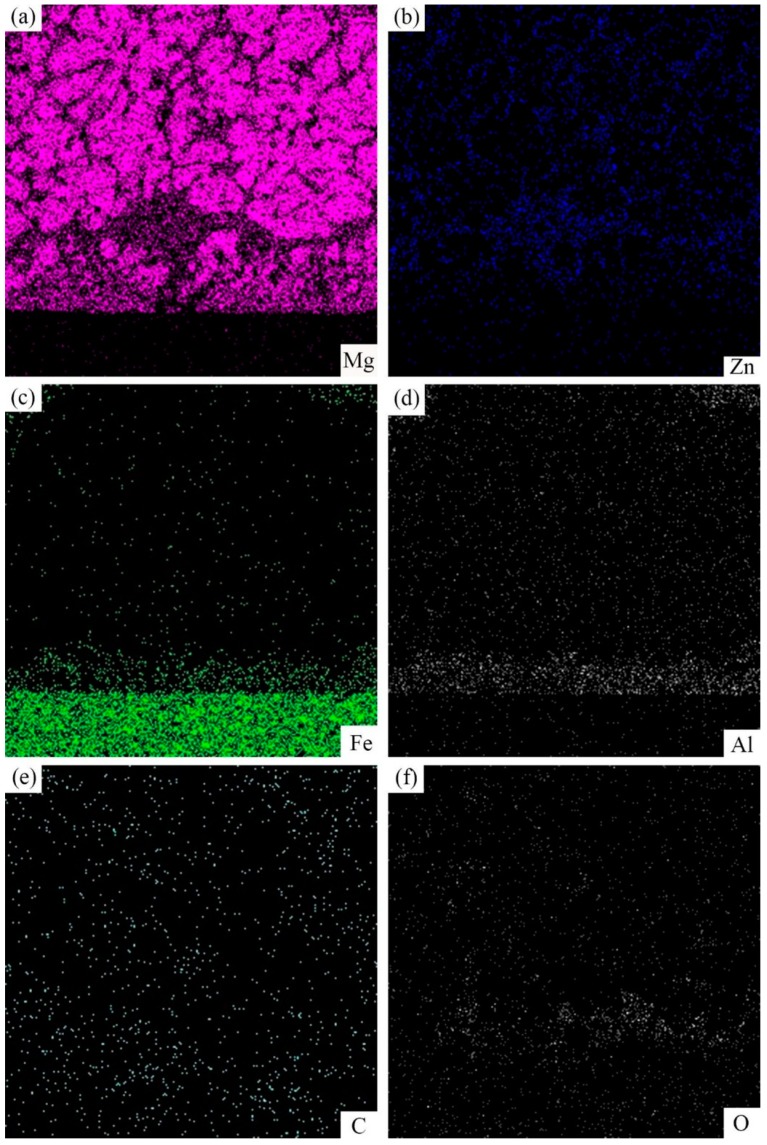
Corresponding EDS maps of the SEM image of the cross-section of galvanized-45 steel/AZ91D bimetallic material. (**a**) EDS map scan of Mg element; (**b**) EDS map scan of Zn element; (**c**) EDS map scan of Fe element; (**d**) EDS map scan of Al element; (**e**) EDS map scan of C element and (**f**) EDS map scan of O element.

**Figure 7 materials-12-01651-f007:**
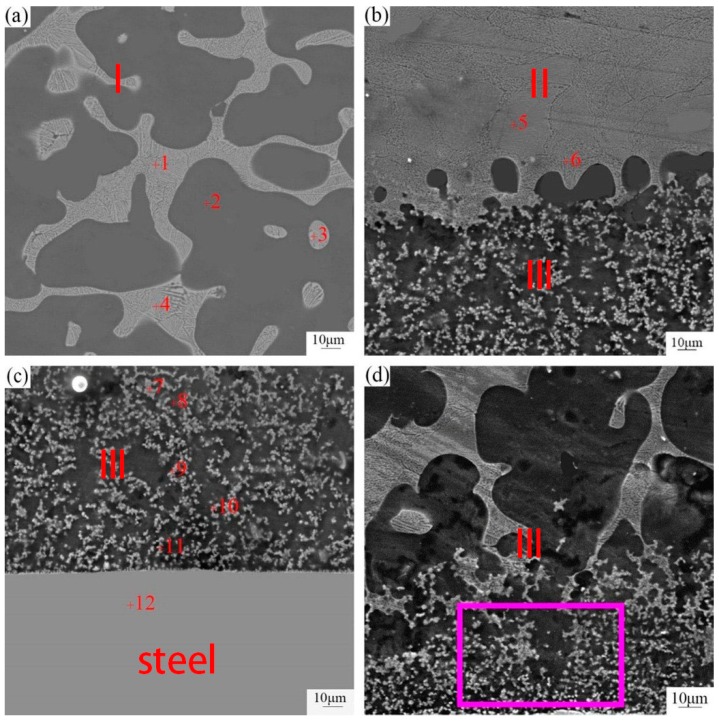
SEM image of the three different layers in galvanized-steel/AZ91D bimetallic material. (**a**) area of layer I in Figure 5a; (**b**) area of the interface between layer I and II in Figure 5a; (**c**) area of the interface between layer III and steel in Figure 5a; and (**d**) area of layer III in Figure 5a.

**Figure 8 materials-12-01651-f008:**
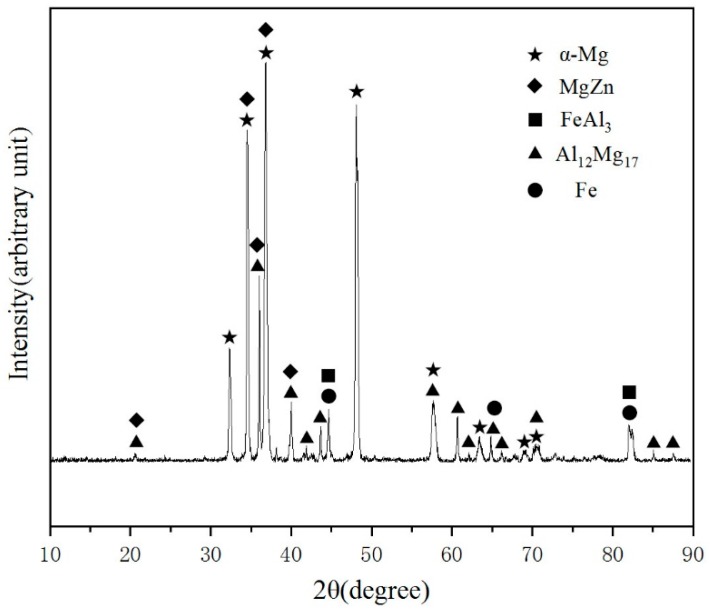
XRD diffraction pattern of the cross-section of galvanized-steel/AZ91D bimetallic material.

**Figure 9 materials-12-01651-f009:**
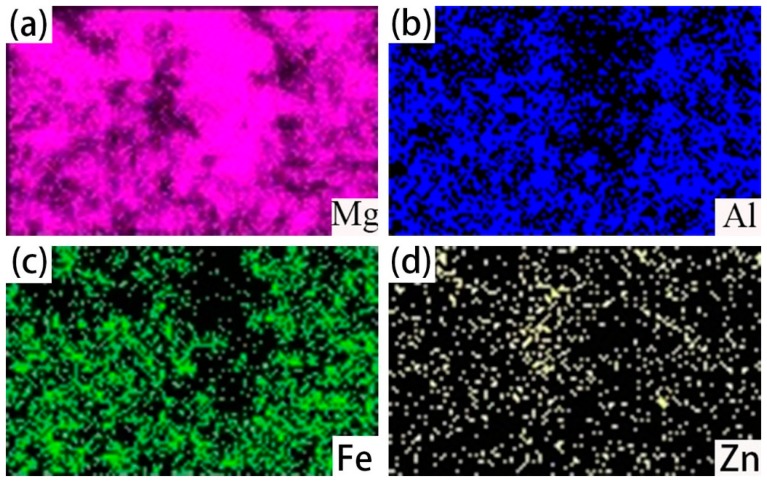
Corresponding EDS maps of the cross-section of the galvanized-steel/AZ91D bimetallic material marked in Figure 7d. (**a**) EDS map scan of Mg element; (**b**) EDS map scan of Al element; (**c**) EDS map scan of Fe element; and (**d**) EDS map scan of Zn element.

**Figure 10 materials-12-01651-f010:**
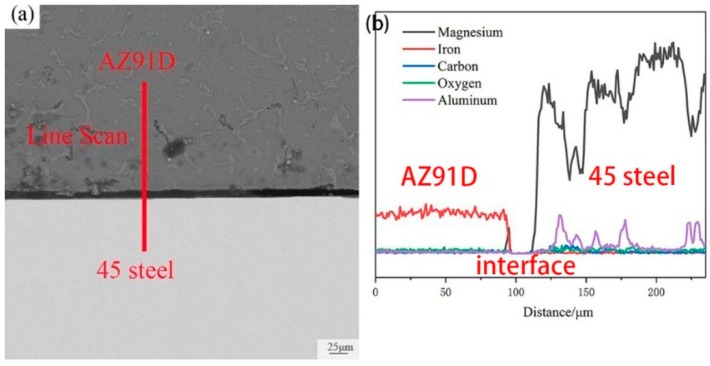
(**a**)The SEM image of the cross-section images of bare-steel/AZ91D bimetallic material; (**b**) EDS line scan of bare-steel/AZ91D interface marked in (**a**).

**Figure 11 materials-12-01651-f011:**
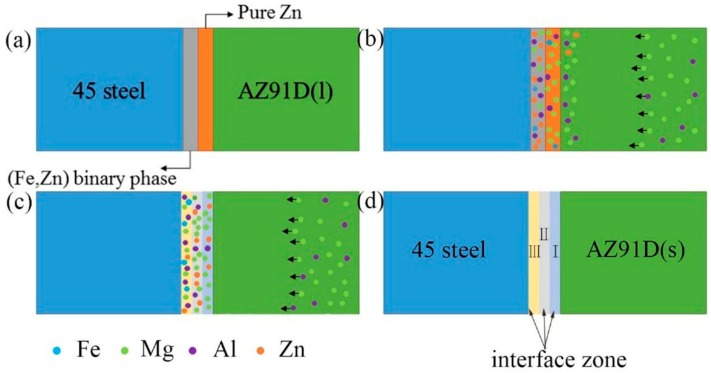
The schematic of microstructure evolution in the galvanized-steel/AZ91D interface. (**a**) filling process of liquid metals; (**b**) diffusion process and chemical reactions; (**c**) metallurgical reaction among the Al, Fe, Mg, Zn elements; and (**d**) solidification.

**Figure 12 materials-12-01651-f012:**
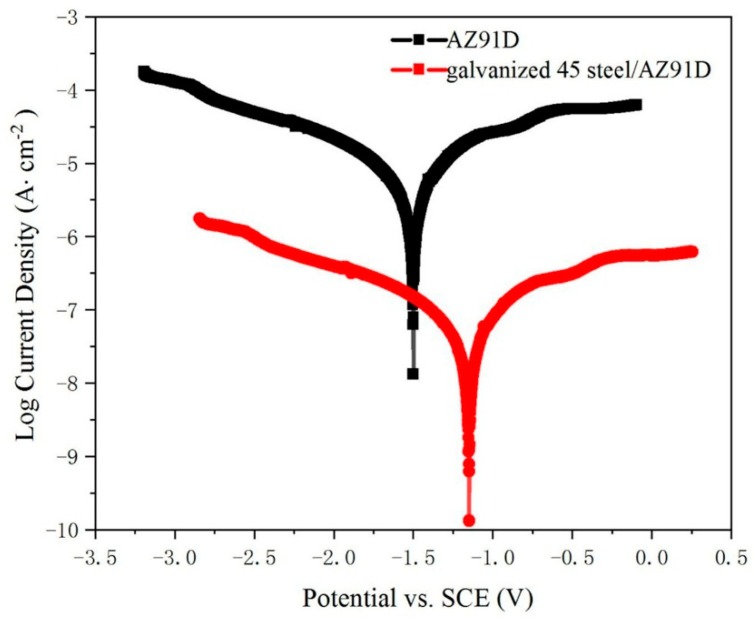
Polarization curves of the galvanized-steel/AZ91D bimetallic materials and AZ91D.

**Figure 13 materials-12-01651-f013:**
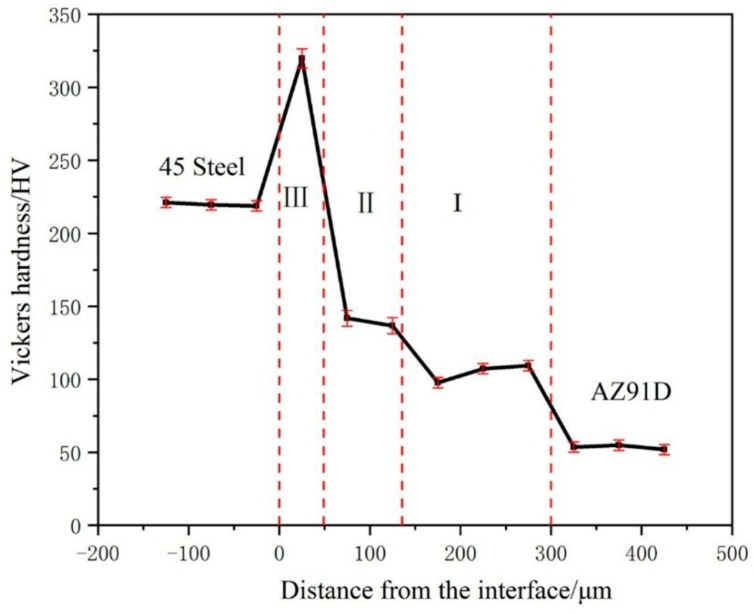
Microhardness distributions at the galvanized-steel/AZ91D compound interface.

**Table 1 materials-12-01651-t001:** Chemical composition (wt.%) of the materials used in this study.

Material	Al	Zn	Mn	Si	Cu	Ni	Cr	S	P	C	Fe	Mg
AZ91D	8.5–9.0	0.45–0.5	0.17–0.4	≤0.05	≤0.02	≤0.01					≤0.004	Bal
45 steel			0.5–0.7	0.17–0.30	≤0.25	≤0.25	≤0.025	≤0.035	≤0.035	0.42–0.50	Bal	

**Table 2 materials-12-01651-t002:** Composition of points 1 to 4 in Figure 3 by EDS analysis.

Point	X(Zn)/%	X(Fe)/%
1	75.47	24.53
2	88.61	11.39
3	92.63	7.37
4	98.61	1.39

**Table 3 materials-12-01651-t003:** Compositions of points 1 to 12 in Figure 7 by EDS analysis.

Point	X(Mg)/%	X(Al)/%	X(Zn)/%	X(Fe)/%	X(C)/%
1	71.36	4.61	24.03	-	-
2	94.46	3.12	2.42	-	-
3	73.45	4.93	21.62	-	-
4	73.04	4.6	22.36	-	-
5	74.53	3.99	21.48	-	-
6	70.32	6.23	23.45	-	-
7	65.34	13.25	19.89	1.52	-
8	92.36	1.37	6.07	0.2	-
9	10.32	58.35	2.33	29.00	-
10	4.83	53.36	3.15	38.65	-
11	4.3	35.5	2.0	58.2	-
12	-	-	-	51.38	48.62

**Table 4 materials-12-01651-t004:** The shear strength of galvanized-steel/AZ91D bimetallic material.

Sample	Shear Strength (MPa)	Average Shear Strength (MPa)
Bare-steel/AZ91D	0	0
Galvanized-steel/AZ91D	12.31	11.81
	13.24	
	9.87	

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
