# Peer review of "Microstructure and Mechanical Properties of Galvanized-45 Steel/AZ91D Bimetallic Material by Liquid-Solid Compound Casting"

_materials, 2019, doi:10.3390/ma12101651_

Round 1
Reviewer 1 Report
Dear Authors,
The subject of the work is very interesting, but the article contains a lot of errors both esential and editorial. Please improve the writing style, introduce numerous corrections and explanation according to the guidelines in the attachment.
Best regards,
Reviewer

Author Response
Point 1: Line 17-18 – Abstract. Eutectics isn’t a phase, it is a two-phase structural component.
Response 1: We thank the reviewer for the precious comments and suggestions. According to the comments from you, some relevant information was inquired and the related issue was modified. The word ‘phase’ is not a right word to modify the ’eutectic’. So we use ‘structure’ to modify eutectic instead of ‘phase’.
Point 2: Line 33 – No reference to Figure 3.
Response 2: We thank the reviewer for the valuable suggestion. The 23th reference in this article was the reference to Figure 3. The related reference symbol was added.
Point 3: Line 43 – Two interlayers (…).
Is the entry in brackets with the word eutectic correct? The composition of eutectic
should be written in round brackets, and maybe the word - eutectics should be
omitted
Response 3: We thank the reviewer for the valuable suggestion. The suggestion was very valuable and we omit the redundant word ’eutectic’.
Point 4: Line 67 – Is this an equilibrium or empirical diagram? There is no explicit
reference in the text.
Response 4: Thank you very much for your valuable advice. The phase diagram I cited was from the reference 6th, which was equilibrium phase diagram.
Point 5: Line 106-107 – Why the testing load was 98g and not 100g?
Is microhardness was determined for the cross-sectional samples?
How many measurements were made for one layer?
Response 5: Thank you so much for your advice. The testing load value was used based on the empirical experiment provide by our teammate.98 g was the most reasonable load for magnesium and magnesium alloy. In order to indicate the hardness measurements more directly, the picture was showed in the below. According to the thickness of different layers, different measurement times were applied. There are three lines in perpendicular to the interface which was employed to measure the microhardness. In every line, different measurement times were applied .The layer close to AZ91D was measured by three times in every line, the layer adjacent to 45 steel was tested by two times and the intermediate layer was measured by one time.
Point 6: How many measurements were made for each of the tests described in the paper?
Response 6: Thanks so much for your advice. Only one specimens was measured to achieve microhardness. And three specimens were measured to achieve shear strength.
Point 7: Line 131-136 – Is this an equilibrium or empirical diagram?
Do you have confidence that this is a reliable equilibrium graph for yours
material? No reference to Figure 6.
Response 7: Thank you for your valuable suggestion. The phase diagram Fig.6 was an equilibrium diagram. But we did not received permission of the Phase Diagrams of ASM Handbook. So we have to delete this Fig.6 and Fig.12. Line 131-136 was analysed based on the Fig.6 and the related equilibrium data.
Point 8: Line 141 – Table 2. Does using an EDS detector for testing the content of carbon guarantees a reliable result?
Response 8: Thank you for your advice. According to the related paper, the EDS detector cannot realize quantitative analysis of light element like carbon. But this result was reliable to qualitative analysis. In this paper, the result by an EDS detector was enough to qualitative analysis. But for the sake of data rigor, we decided to delete this date for point 1 in this Fig.4 and Table2.
Point 9: Line 148 – Is this an equilibrium or empirical diagram?
There is no explicit reference in the text.
Do You have confidence that this is a reliable equilibrium graph for yours
material?
Response 9: Thank you for your valuable suggestion. The phase diagram Fig.6 was an equilibrium diagram. But we did not received permission of the Phase Diagrams of ASM Handbook. So we have to delete this Fig.6 and Fig.12. And this graph was cited by 26th.
Point 10:Line 178 – The phase composition is given. The data presented in Figure 10 do not confirm this unambiguously.
Response 10: Thanks for your advice. Because the dimension of XRD point test was much bigger than the interface, it could not realize that XRD point test is applied in each zone of the layer and the phase composition are compared. In the experience, the XRD point was posited into the center of interface zone. In addition, in order to achieve more correct result, the scan rate was so slow at 4 degree/min range from 10 degree to 90 degree. In our paper, the XRD was applied to identify the phase composition of Fe/Mg bimetallic material including 45 steel , interface zone and AZ91D. And this result was enough to prove the correct of the result which was achieved by the EDS point scan and the related phase diagram.
Point 11: Line 182 – Eutectic isn’t a phase, but a multi-phase structural component.
Response 11: Thanks for your advice. Eutectic is not a phase and we use the structure instead of phase.
Point 12: Line 200 – On what basis has the composition of particular phases been
determined? It doesn’t follow unequivocally from the Figure 10.
Response 12:Thanks for your advice. According to the result of Figure 10, the existence of α-Mg, FeAl3, MgZn can be proved. And Al12Mg17 and Fe may exist into the 45 steel and AZ91D, respectively. According to the reference (7-10) and binding energy, the Fe atom prefer to react with Al and Mg prefer to react with Zn. So it can be concluded that the Al12Mg17 and Fe do not exist in the interface zone.
Point 13: Line 207 – Is this an equilibrium or empirical diagram?
There is no explicit reference in the text.
Response 13: Thank you for your valuable suggestion. The phase diagram Fig.6 was an equilibrium diagram. But we did not received permission of the Phase Diagrams of ASM Handbook. So we have to delete this Fig.6 and Fig.12. And this graph was cited by 26th.
Point 14: . Line 228 – The diffusion is a process, not a reaction, like a chemical reaction for example.
Response 14 : Thank you so much for your advice. The diffusion is a process, not a reaction. So we use process to modify diffusion instead of reaction.
Point 15:Line 236,238 – Eutectic isn’t a phase, but a multi-phase structural component.
Response 15: Thank you so much for your advice. Eutectic is not a phase and we use structure to modify the eutectic instead of phase
Point 16: Line 256-257 and 259 – The given values of potentials do not correspond with the graph in Fig. 15, and more precisely with the description on the x-axis and y-axis.
Response 16 : Thank you so much for your advice. The given values of potentials do not correspond with the graph in Fig.5. And we changed the concept of x axis and y axis in Fig.15.
Point 17: Line 259 – Please, verify the numerical values and units with reference to the
graph in Figure 15.The given values of potentials do not correspond
Response 17:Thanks for your valuable advice. According to your comments, the mistake of numerical values and units in the text was modified.
Point 18: Line 268 – Please, verify the chart with reference to the text.
Response 18:Thanks for your valuable advice According to your comments, we modified the corresponding error in text.
Point 19:Line 273-279 – You write about the average values of microhardness. How many microhardness measurements were made for individual average values?
Response 19: Thanks for your valuable suggestions. In order to indicate the hardness measurements more directly, the picture was showed in the below. According to the thickness of different layers, different measurement times were applied. There are three lines in perpendicular to the interface which was employed to measure the microhardness. In every line, different measurement times were applied .The layer close to AZ91D was measured by three times in every line, the layer adjacent to 45 steel was tested by two times and the intermediate layer was measured by one time.
Point 20:Line 282 – It's a possible average value from one measurement (concerns I layer on Figure 16)?
Response 20:Thanks for your valuable advice. The layer marked in the Figure was not corresponded with the layer in the text. So, we modified the related information in the text.
21. Line 297-299 – It is rather a summary, not a conclusion.
Response 21: Thankes for your valuable advice. According to your comments, the relevant information has been modified as follows:
Original text: (1) The hot-dip galvanizing coating was generated on the surface of 45 steel at 500 °C for 120s. And after preheating (at 200 °C for 60 minnuite) and pouring, galvanized 45 steel/AZ91D metallic material was achieved by compound casting.
Modified text: (1) Metallurgical bonding was achieved between AZ91D and galvanized 45 steel by solid-liquid compound casting, while there is a gap between AZ91D and bare 45 steel by solid-liquid compound casting.

Reviewer 2 Report
Abstract - Lack of a clearly defined objective.
Introduction - literature e.g. 1-3 not very modern. Only 1 literary item from 2018. Please complete this.
„The reliable combination between steel and Mg alloy was an effective method to realize the better operational performance.” Where this combination was specifically used?
Methodology - On the basis of which standards mechanical tests were carried out?
Figures 1 and 2 are simple but unclear for ordinary readers.
Results – „The average 122 thickness of Zn coating was 28um” What method was used to measure the thickness of Zn. Please use two methods to confirm the results. Complete the methodology.
Fig. 8. Oxygen was found in the material. What is the influence of oxygen on the properties of hot-169 dip galvanized 45 steel/AZ91D bimetallic material? Please explain it.
Fig. 10. Where on the cross-section of the sample was XRD measured? The XRD point test should be applied in each zone of the layer and the phase composition should be compared. Complete the methodology.
Complete the surface roughness tests. This is an important feature of the layers.
The formation mechanism of the phases including intermediate compound and solid solution at the galvanized 45 steel/AZ91D interface present please in the scheme. Line 224-241.
„The schematic of microstructure evolution in the galvanized 45 steel/AZ91D interface.” Is presented in the Fig. 14.
hardness measurement -
On what basis was the load selected?
How many hardness measurements were made? It seems that single measurements were taken. Please calculate the measurement error.
In the whole text there are sporadic typos and no spaces, e.g. line 101, 106, 107, ….
The graphs are too large, unprofessional made. The quality of the figures is a cause for concern.
Author Response
Point1:Abstract - Lack of a clearly defined objective.
Response1: Thank you very much for your valuable advice. According to your comments, the clearly defined objective was added and the sentence was showed in the below:
‘The connection between hot-dip galvanized 45 steel and AZ91D was achieved by liquid-solid compound casting to realize the light weight.’
Point 2:Introduction - literature e.g. 1-3 not very modern. Only 1 literary item from 2018. Please complete this.
Response2: Thank you very much for your valuable advice. In the paper, three modern literature with high citation was cited to replace the literature 1-3 :
[1] A.A. Luo, Magnesium casting technology for structural applications, Journal of Magnesium and Alloys, 1 (2013) 2-22.
[2] T.E. B.L. Mordike, Magnesium — properties–applications–potential, Materials Science and Engineering a-Structural Materials Properties Microstructure and Processing, (2001) 37-45.
[3] KULEKCI M K, Magnesium and its alloys applications in automotive industry.The International Journal of Advanced Manufacturing Technology, 39(2007): 851-865.
Point 3:“The reliable combination between steel and Mg alloy was an effective method to realize the better operational performance.” Where this combination was specifically used?
Response3: First of all, thanks so much for your valuable advice. According to the comments from you and the editors, we carefully modified this sentence with specifically used. The reliable combination between steel and Mg alloy was an effective method to realize light weight with better operational performance in automobile field
Point 4:Methodology - On the basis of which standards mechanical tests were carried out?
Response4: We thank the reviewer for the precious comments and suggestion. According to your comments form you, some detailed standards mechanical test were carried out in the article. The microhardness standards was GB/T4340. 1 —1999, which was used to measure the vickers hardness of metals and the shear strength standards was CSN 66 8510-1960 which was employed to measure shear adhesive strength under tension load.
Point 5:Figures 1 and 2 are simple but unclear for ordinary readers.
Response 5: Thanks for your valuable advice. According your comments from you, Figures 1 was replaced by more detailed picture in the below. In addition, the Figure 2 was carefully modified and showed in the text.
Point 6:Results – „The average 122 thickness of Zn coating was 28um” What method was used to measure the thickness of Zn. Please use two methods to confirm the results. Complete the methodology.
Response 6:Thanks for your valuable advice. The thickness of zinc coating was measured by two different methods( gravimetric thickness gauge and EDS scan ). Firstly the gravimetric thickness gauge methods was applied. During this measurement process, the weight of 45 steel with and without zinc coating and the superficial area of the zinc coating surface were measured by accuracy balance. And according to the equal in the below, the thickness of zinc coating was achieved.
T=(m×A)/(A×d)
Where T represents thickness,unit: μm;m represents the weight of the coating, unit :mg;A represents the superficial area of the zinc coating surface,unit: g/cm2;d present the density,unit :g/cm3。
Second methods was EDS scan. According to the EDS map scan and line scan, the area and thickness in the picture of zinc coating was achieved, which was deduced by the distribution of zinc element. The thickness of zinc coating in the picture divide the scale length of 10μm time 10 is equal to the thickness of zinc coating.
Point 7:Fig. 8. Oxygen was found in the material. What is the influence of oxygen on the properties of hot-dip galvanized 45 steel/AZ91D bimetallic material? Please explain it.
Response 7: Thanks for your valuable advice. According to the comments from you, the influence of oxygen on the properties of hot-dip galvanized 45 steel/AZ91D bimetallic material was expounded in the paper .The sand blasting was applied to break the oxidation layer on the surface of zinc coating , so in this condition, the O element was negligible. O element is mainly formed in the layer close to steel. So, O element may react with Mg and Zn to generate impurity which can easily result in the crack during compound casting. In our article, O element was negligible and may exist as MgO or ZnO.
Point 8:Fig. 10. Where on the cross-section of the sample was XRD measured? The XRD point test should be applied in each zone of the layer and the phase composition should be compared. Complete the methodology.
Response 8:Thanks for your valuable advice. Because the dimension of XRD point test was much bigger than the interface, it could not realize that XRD point test is applied in each zone of the layer and the phase composition are compared. In the experience, the XRD point was posited into the center of interface zone. In addition, in order to achieve more correct result, the scan rate was so slow at 4 degree/min range from 10 degree to 90 degree.
Point 9:Complete the surface roughness tests. This is an important feature of the layers.
Response 9: Thanks for your valuable advice. According to your comments ,the zinc-coating surface roughness tests was applied by roughness tester SV-2100.The value of the surface roughness tests reached Ra1.1μm.The variation of the surface roughness was little because of the smooth surface.
Point 10:The formation mechanism of the phases including intermediate compound and solid solution at the galvanized 45 steel/AZ91D interface present please in the scheme. Line 224-241.
„The schematic of microstructure evolution in the galvanized 45 steel/AZ91D interface.” Is presented in the Fig. 14.
Response 10: Thanks for your valuable advice. According to your comments, we modified the sentence “The formation mechanism of the phases including intermediate compound and solid solution at the galvanized 45 steel/AZ91D interface” to the schematic of microstructure evolution in the galvanized 45 steel/AZ91D interface.
Point 11:hardness measurement -
On what basis was the load selected?
How many hardness measurements were made? It seems that single measurements were taken. Please calculate the measurement error.
Response 11 :Thanks for your valuable suggestions. In order to indicate the hardness measurements more directly, the picture was showed in the below. According to the thickness of different layers, different measurement times were applied. There are three lines in perpendicular to the interface which was employed to measure the microhardness. In every line, different measurement times were applied .The layer close to AZ91D was measured by three times in every line, the layer adjacent to 45 steel was tested by two times and the intermediate layer was measured by one time.
Point 12:In the whole text there are sporadic typos and no spaces, e.g. line 101, 106, 107, ….
Response 12: Thanks for your hard work. According to your comments, we carefully checked the whole text and modified some mistakes such as sporadic typos and no space.
Point 13:The graphs are too large, unprofessional made. The quality of the figures is a cause for concern.
Response 13: Thanks for your valuable advice. According to your comments, the size of the graphs was changed and the quality was enhanced.

Reviewer 3 Report
The paper is interesting but there are some comments:
Please define the material: 45 steel, HSLA. and also other terms, such as RJ-2 flux
The abstract should be rewritten with numerical data, not only ”better shear strength” or ”much better”
The English language has to be checked in rows: 14, 20, 31,38, 56,120 and so on...
I suggest reconsidering Fig 2 and paragraph 2.2 in a more clearer form.
The testing load should be given as a force, in N.
Figs 6, 12 are not presented distinctively in the paper, only as a reference (26)
In my opinion, the signs in Fig 10 are not appropriate for a research paper.
The conclusions must be seriously improved.
Author Response
Point 1: Please define the material: 45 steel, HSLA. and also other terms, such as RJ-2 flux.
Response 1: We thank the reviewer for the precious comments and suggestions. 45 steel was one standard brand in China. In ASTM, 45 steel was equal to 1045, and in JIS, 45 steel was equal to S45C. So, 45 steel is the useful definition. HSLA was Referred to as High Strength Low Alloys which was one of ASTM.RJ-2 flux is one of the most popular coating agent of magnesium alloys in China. If you have some question about it, please reply to me. Thanks a lot!
Point 2: The abstract should be rewritten with numerical data, not only ”better shear strength” or ”much better”
Response 2: We thank the reviewer for the precious comments and suggestions. According to the comments from you, some numerical data was added into the suitable position in the paper and the typical modification was showed in the below:
1. the adhesive strength of galvanized 45 steel and AZ91D was improved to 11.81MPa.
2. In addition, the corrosion resistance of galvanized 45 steel/AZ91D bimetallic material was much better than AZ91D that corrosion potential increased from -1.493V to -1.143V and corrosion current density changed from 3.015´10-5 A/cm2 to 1.34´10-7 A/cm2.
Point 3: The English language has to be checked in rows: 14, 20, 31,38, 56,120 and so on...
Response 3:Thanks for your valuable advice.According to your comment, the English language was check by me carefully and the relvant mistake was modified.
Point 4: I suggest reconsidering Fig 2 and paragraph 2.2 in a more clearer form.
Response 4:We thank the reviewer for the precious comments and suggestions. According to your comment, Fig.2 was modified carefully to reflect the description in the paragraph 2.2 fully. The process of pre-treatment was added before hot-dip and the process of hot-dip was modified in order to present the process carefully.
Point 5:The testing load should be given as a force, in N.
Response 5: We thank the reviewer for the precious comments and suggestions. According to your comment , some articles related to microhardness test were scan. And in a conclusion, the ‘g’ is one of effective method to represent the testing load.
But ‘N’ was easier to undertand the applied load. The related information was change in the article (50g-50N,98g-98N)
Point 6: Figs 6, 12 are not presented distinctively in the paper, only as a reference (26)
Response 6: Thanks for your valuable advice. According to your comment, the Figure 6 and 12 are not presented distinctively in this paper, but I want to use some sentences in the below paragraph to show different opinion.
Fig.6 shows SEM micrograph of interfacial microstructure of hot-dip galvanized 45 steel/AZ91D interface and the related EDS line scan.Fig.6 is one important evidence to represent the interfacial microstructure and the related element distribution between 45 steel and AZ91D. If there is no Fig.6 in the article, there are not enough to confirm the metallurgical bonding between galvanized 45 steel and AZ91D. So, it can be concluded that Fig.6 is presented distinctively in the paper.
Fig.12 shows the schematic of microstructure evolution in the galvanized 45 steel/AZ91D interface which was the important evident to illustrate the schematic of microstructure evolution. Combination the statement in the article and Fig.12, the bond mechanism was understand fully. Bonding Mechanism of galvanized 45 steel/ AZ91D can be fully understood at the first sight of the picture. In order to explain the bond mechanism, the relevant modification was showed in the below:
Based on the above analysis, the schematic illustration of formation was showed in the below and can be divided into four stages: (1) filling process of liquid metals (2) diffusion process and chemical reactions; (3) metallurgical reaction among the Al, Mg , Zn elements; (4)solidification as showed in Fig.12
Point 7: In my opinion, the signs in Fig 10 are not appropriate for a research paper.
Response 7: Thanks for your valuable advice.According to your comment, the signs in Fig.10 and Fig.9 was modified in the article.
Point 8: The conclusions must be seriously improved
Response 8:Thanks for your valuable advice. According to your comment, the conclusions has been modified carefully.
Conclusion:
(1) In hot-dip galvanized 45 steel/AZ91D bimetallic material ,based on the existence of the interface zone, the metallurgical bonding between hot-dip galvanized 45 steel and AZ91D was achieved via pouring the molten magnesium alloy into the mould inserted into galvanized 45 steel. However, bare 45 steel/AZ91D bimetallic material was achieved by liquid-solid compound casting due to mechanical bond instead of metallurgical bonding based on the existence of a gap between bare 45 steel and AZ91D.
(2) The interface zone between galvanized 45 steel and AZ91D can be divided into three different layers. The layer adjacent to the AZ91D (layer I) was mainly composed of (α-Mg + MgZn) eutectic structure and black block phase (α-Mg). The layer close to the 45 steel (layer III) was mainly comprised of small white block FeAl3 and black block α-Mg. And the intermediate layer(layer II) was white uniform lamellae phase (α-Mg + MgZn) eutectic structure.
(3) The hot-dip galvanized 45 steel on the surface of galvanized 45 steel/AZ91D bimetallic material could efficiency improve the corrosion resistance of AZ91D that corrosion potential increased from -1.493V to -1.143V and corrosion current density changed from 3.015´10-5 A/cm2 to 1.34´10-7 A/cm2.
(4) With the change of the composition in different layers, the microhardness of galvanized 45 steel/AZ91D bimetallic material varies from location to location. From the layer I to the layer II, the microhardness increased gradually from 104.8HV to 139.3HV due to the increasement of MgZn phase contents. But from the layer II to the layer III, the microhardness changes rapidly from 139.3HV to 325.4HV, because the microhardness of the FeAl3 was much larger than the MgZn phases.
(5) The shear strength which reach 11.81MPa of galvanized 45 steel/AZ91D bimetallic material was much better than bare 45 steel/AZ91D bimetallic material ,because metallurgical bond replace mechanical bond.

Reviewer 4 Report
This manuscript includes interesting results on the joining of steel and magnesium alloy by compound casting. However, there are several points need to be revised, as be listed below.
(1) There are so many mistakes in English on spelling, tense, grammar, missing period, upper and lower case letters, etc. Check by a native speaker is strongly recommended.
(2) In this manuscript, references larger than or equal to [23] are not listed. Please check again.
(3) It is hard to understand the purposes of this work. The specification of the purposes in "Introduction" is quite important to evaluate the originality of this work.
(4) Fig. 1: Is the title of horizontal axis in Fig. 1 correct? In Fig. 1(b), the unit of temperature is K in the vertical axis but °C in the figure.
(5) Figs. 2 and 3: Please add dimensions to the important parts and edges.
(6) Fig. 4: It is impossible to find the boundaries among layers A to D, and also the points 1~4. The origin (distance equals zero) of the line scan is not specified.
(7) Fig. 6: It is hard to distinguish layer II in Fig. 6. The authors tell that layer II consists of white phase, but is it really a layer?
(8) The range of each layer should be denoted in Fig. 6 (b), Fig. 7, and Fig. 8 to help readers' understanding.
(9) Fig. 12: Authors show the schematic of microstructure evolution in Fig. 12 but there is no explanation about it. The formation of three different layers should be explained by referring Fig. 12.
(10) Table 4: It seems strange to show the zero shear strength in Table 4. Does it have any contribution to the purpose of this work?
(11) Since the different load and the different holding time were applied between AZ91D and 45 steel in the microhardness test, is it reasonable to directly compare Vickers hardness in Fig. 14?
(12) Supposing the purpose of this study, the discussion on relationships between the hardness and the shear strength with taking into account the microstructure is necessary.
(*) "MPa" not "Mpa".
(*) P. 3, Lines 110 and 111: "g" is not the unit of load.
(*) P. 5, Line 157: What is "A356"?
Round 2
Reviewer 1 Report
Dear Authors,
changes that have been made allow you to submit work for publication, but the work requires minor editing language.
Best regards,
Reviewer
Author Response
Point: changes that have been made allow you to submit work for publication, but the work requires minor editing language.
Response: We thank the reviewer for the precious comments and suggestions. According to the comments from you, we carfully examine the paper with many corrections which are indicated in yellow background in the revised manuscript

Reviewer 2 Report
Please correct:
Figure 1. Mg-Fe binary phase diagram [6]. Phase diagram ?
Fig. 6. (a)the, material;(b) - spaces
Author Response
Point 1: Figure 1. Mg-Fe binary phase diagram [6]. Phase diagram ?
Response 1: Thank you for your valuable suggestion. I am so sorry to make this mistake to add these needless phrase ‘Phase diagram’.So i decide to delete this needless phrase.
Point 2:Fig. 6. (a)the, material;(b) - spaces
Response 2: We thank the reviewer for the precious comments and suggestions. According to the comments from you, I decide to modify this sentence in Fig.6 and the related paper. The relevant information has been modified as follows:
Original text: Fig. 6. (a)the Scanning Electron Micrograph (SEM) of the cross-section images of hot-dip galvanized 45 steel/AZ91D bimetallic material;(b) EDS line scan of galvanized 45 steel/AZ91D interface marked in (a).
Modification :(a) =SEM micrograph of interfacial microstructure of hot-dip galvanized 45 steel/AZ91D interface (b) =EDS line scan of hot-dip galvanized 45 steel/AZ91D interface marked in (a).
Original text: Fig.6 shows the result from SEM of the cross-section of AZ91D/galvanized 45 steel and the related EDS line scan.
Modification: Fig.6 shows SEM micrograph of interfacial microstructure of hot-dip galvanized 45 steel/AZ91D interface and the related EDS line scan.

Reviewer 3 Report
I consider that the authors answered to all my suggestions and comments.
Author Response
Response to Reviewer 3 Comments
Cover letter
Dear reviewer:
We submit our manuscript entitled “Microstructure and mechanical properties of hot-dip galvanized 45 steel/AZ91D bimetallic material by liquid-solid compound casting” to ‘Materials’. We believe that the following aspects of this manuscript will make it interesting to you. We used the liquid-solid compound casting technology to realize the combination between galvanized 45 steel and AZ91D. Compared with bare 45 steel/AZ91D bimetallic material, the galvanized 45 steel/AZ91D bimetallic material had better shear strength because metallurgical bonding replaced mechanical bond. In addition, from 45 steel to AZ91D the micro hardness of the galvanized 45 steel/AZ91D bimetallic material changed due to the variation of the composition. In comparison to AZ91D, galvanized 45 steel/AZ91D bimetallic material had better corrosion resistance.
Thank you very much for your attention and consideration.
Sincerely yours,
Jun Cheng
Reply to the reviews
First of all, we would like to thank you for reviewing our manuscript very carefully and putting forth your constructive comments. In order to enhance the quality of the paper, some English check was applied. Changes, including minor ones not related to the reviewers’ comments, are indicated in red colour in the revised manuscript.

Reviewer 4 Report
Thank you for your sincere answers to my comments. I think the revised manuscript is suitable for publication in the journal "Materials". Only, I would like to recommend English check again.
Author Response
Response to Reviewer 4 Comments
Cover letter
Dear reviewer:
We submit our manuscript entitled “Microstructure and mechanical properties of hot-dip galvanized 45 steel/AZ91D bimetallic material by liquid-solid compound casting” to ‘Materials’. We believe that the following aspects of this manuscript will make it interesting to you. We used the liquid-solid compound casting technology to realize the combination between galvanized 45 steel and AZ91D. Compared with bare 45 steel/AZ91D bimetallic material, the galvanized 45 steel/AZ91D bimetallic material had better shear strength because metallurgical bonding replaced mechanical bond. In addition, from 45 steel to AZ91D the micro hardness of the galvanized 45 steel/AZ91D bimetallic material changed due to the variation of the composition. In comparison to AZ91D, galvanized 45 steel/AZ91D bimetallic material had better corrosion resistance.
Thank you very much for your attention and consideration.
Sincerely yours,
Jun Cheng
Reply to the reviews
First of all, we would like to thank you for reviewing our manuscript very carefully and putting forth your constructive comments. In order to enhance the quality of the paper, some English check was applied. Changes, including minor ones not related to the reviewers’ comments, are indicated in red colour in the revised manuscript.
